Improved ultrastructure of marine invertebrates using non-toxic buffers

Montanaro Jacqueline 1
Gruber Daniela 2
Leisch Nikolaus 3 4 nleisch@mpi-bremen.de
1 OCUVAC—Center of Ocular Inflammation and Infection, Laura Bassi Centers of Expertise, Center for Pathophysiology, Infectiology and Immunology, Medical University of Vienna , Vienna , Austria
2 Core Facility Cell Imaging and Ultrastructure Research, University of Vienna , Vienna , Austria
3 Max Planck Institute for Marine Microbiology , Bremen , Germany
4 Department of Ecogenomics and Systems Biology, University of Vienna , Vienna , Austria
Reimer James
Electronic publication date: 2016 Mar 31
Publication date: 2016
Volume: 4
Electronic Location ID: e1860
Received 2016 Jan 11; Accepted 2016 Mar 9
Copyright: ©2016 Montanaro et al.
Copyright year: 2016
Copyright holder: Montanaro et al.
License: This is an open access article distributed under the terms of the Creative Commons Attribution License, which permits unrestricted use, distribution, reproduction and adaptation in any medium and for any purpose provided that it is properly attributed. For attribution, the original author(s), title, publication source (PeerJ) and either DOI or URL of the article must be cited.
License URL: https://creativecommons.org/licenses/by/4.0/

Keywords: Electron microscopy, Immersion fixation, PHEM buffer

Funding: Max Planck Society European Research Council Advanced Grant 340535 Austrian Science Fund P22470-B17 Austrian Research Promotion Agency 822768 Republic of Austria NL was supported by the Max Planck Society, the ERC Advanced Grant 340535 and the Austrian Science Fund (FWF) grant P22470-B17. JM was funded by the “Laura Bassi Centers of Expertise” Program of the Austrian Federal Ministry of Economy through the Austrian Research Promotion Agency (FFG Project Number: 822768, PI Talin Barisani-Asenbauer) and the Republic of Austria. The funders had no role in study design, data collection and analysis, decision to publish, or preparation of the manuscript.

==============================
Many marine biology studies depend on field work on ships or remote sampling locations where sophisticated sample preservation techniques (e.g., high-pressure freezing) are often limited or unavailable. Our aim was to optimize the ultrastructural preservation of marine invertebrates, especially when working in the field. To achieve chemically-fixed material of the highest quality, we compared the resulting ultrastructure of gill tissue of the mussel Mytilus edulis when fixed with differently buffered EM fixatives for marine specimens (seawater, cacodylate and phosphate buffer) and a new fixative formulation with the non-toxic PHEM buffer (PIPES, HEPES, EGTA and MgCl2). All buffers were adapted for immersion fixation to form an isotonic fixative in combination with 2.5% glutaraldehyde. We showed that PHEM buffer based fixatives resulted in equal or better ultrastructure preservation when directly compared to routine standard fixatives. These results were also reproducible when extending the PHEM buffered fixative to the fixation of additional different marine invertebrate species, which also displayed excellent ultrastructural detail. We highly recommend the usage of PHEM-buffered fixation for the fixation of marine invertebrates.

Introduction

Marine research is often dependent upon species sampling from off-shore research stations, marine vessels and submersibles. As highlighted by the Census of Marine Life, many species remain undiscovered, while the complex details about many others remain unknown (Census of Marine Life, 2010). One of the routine techniques of marine biology research is to preserve sample specimens for light and electron microscopy, for example for the formal description of a new species. Ultrastructural research is especially valuable for elucidating details on the symbiotic relationships between larger metazoan and prokaryotic organisms, like the mussels of the genus Bathymodiolus. These mussels live at hydrothermal vents and cold seeps in the deep sea (reviewed in Dubilier, Bergin & Lott, 2008). The mussels harbor chemoautotrophic bacterial symbionts in their gills which exploit the fluid chemistry at these sites to fix carbon and sustain their host (reviewed in Petersen & Dubilier, 2009).

The first and most crucial step for successful ultrastructure analysis is the fixation of the specimen to preserve the morphology of cells with minimal alteration from the living state (Hayat, 2000). There are currently two methods regularly used for sample fixation, high pressure freezing and chemical fixation. High-pressure freezing relies on extremely rapid cooling to vitrify the water in the sample and is usually followed by dehydration at ultra-low temperatures (freeze-substitution) and infiltration with resins (Kuo, 2014). Chemical immersion fixation is conventionally based on aldehydes such as glutaraldehyde (GA) or formaldehyde (FA) or a combination of both, which cross-link proteins (Dykstra & Reuss, 2003; Hayat, 2002). It is followed by stepwise dehydration and infiltration with resins. Due to its convenience, low cost and availability, it remains the most widely used method for preserving biological specimens for electron microscopy. Cellular components and ultrastructural details are adequately preserved, whilst the technique itself is easy to apply and requires minimal equipment and expertise (Hayat, 2000). Additionally, when working on research vessels or remote research stations, access to techniques like high-pressure freezing is either extremely limited or nonexistent.

Regardless of which fixative is used, any artifact or structural changes introduced during the fixation step (e.g., due to changes of pH or osmolarity), cannot be corrected in later stages and may lead to poor ultrastructure preservation. Therefore, the aldehydes are applied with a buffer, which needs to act as solvent for the fixative, maintain a specific pH and convey tonicity to the final fixative solution. The most commonly used buffers for ultrastructure fixation are cacodylate buffer and phosphate buffer (Dykstra & Reuss, 2003). As an adaption for fixation of marine invertebrates, sometimes diluted seawater is used (Dykstra & Reuss, 2003; Ettensohn, Wray & Wessel, 2004). All of the above buffers come with trade-offs; seawater is, by nature, isotonic to marine samples but has little buffering capacity. Phosphate buffer was reported to cause precipitation artifacts in the tissue (Hayat, 2000; Przysiezniak & Spencer, 1989) and cacodylate buffer contains arsenic and can have a toxic effect on the sample prior to fixation, which can alter membrane permeability and affect subcellular preservation. Additionally, arsenic gas can be produced in presence of acids, posing a health hazard. According to the Globally Harmonized System of Classification and Labeling of Chemicals, it must be disposed of as hazardous waste (Electron Microscopy Sciences, 2015). Some toxic components are essential for electron microscopy (e.g., the fixative for immersion fixation) however, there has been a concerted effort to reduce the toxic materials used (e.g., replacing uranyl acetate with either gadolinium or samarium (Nakakoshi, Nishioka & Katayama, 2011)).

The non-toxic PHEM buffer has a wide pH range, good buffering capacity and causes no precipitations with any reagents used during sample processing. It is a combination of the two zwitterionic chemicals PIPES and HEPES with EGTA and MgCl2 and was proposed by Schliwa & Van Blerkom (1981). HEPES seems to stabilize the lipid components of cell membranes and PIPES causes retention of cellular material, reduces lipid loss in the cells and facilitates extensive cross-linking of cellular material (Baur & Stacey, 1977; Hayat, 2000). The addition of EGTA, a chelating agent with a high affinity for calcium ions, as well as magnesium chloride enhances the preservation of microtubules and membranes. Therefore, PHEM would seem to be an ideal electron microscopy buffer. However until now, its traditional use has been limited to extraction stabilization of eukaryotic cytoskeleton (Schliwa & Van Blerkom, 1981), immunofluorescence applications (in e.g., Dictyostelium discoideum (Koonce & Gräf, 2010), embryos of Danio rerio (reviewed in Schieber et al., 2010)) as well as immuno-electron microscopy (in e.g., Saccharomyces cerevisiae (Griffith et al., 2008)) of either single cell organisms or cell culture monolayers.

The aim of this study was to compare the effect different buffers have on the ultrastructural preservation of marine invertebrates and explore the usage of PHEM buffer in combination with glutaraldehyde. We measured the osmolarity of each of the different buffers and fixatives and adapted the concentration of the PHEM buffer to formulate a new isosmotic buffer-fixative combination. This formulation was compared to established buffer- fixative combinations using the gill tissue of the marine invertebrate Mytilus edulis, due to its ready availability. After evaluation of the initial experiment, the PHEM buffered fixative was applied to the fixation of the symbiotic deep-sea mussel Bathymodiolus childressi.

Methods

Buffer and fixative preparation and osmolarity measurements

A 10X stock solution of the PHEM buffer was prepared according to (Schliwa & Van Blerkom, (1981)) by dissolving 600 mM PIPES, 250 mM HEPES, 100 mM EGTA and 20 mM MgCl2 in 100 ml of ddH2O. The pH was raised above 7.0 with 10M KOH for all components to fully dissolve. Final pH was adjusted to 7.4.

A 10X PBS stock solution (pH 7.4) was prepared by dissolving 137 mM NaCl, 2.7 mM KCl, 10 mM Na2HPO4 and 2 mM KH2PO4 in 1 l of ddH2O. 0.2M Sodium cacodylate buffer (pH 7.4) and 25% glutaraldehyde were obtained from Scientific Services, Germany. Unless otherwise indicated, all components were purchased from Carl Roth, Germany.

Fixative solutions were prepared according to Table 1. All fixatives were prepared from the same stock of 25% GA and contained a final concentration of 2.5% GA. The osmolarity of the fixative is usually adjusted using non-electrolytes like sucrose, glucose or dextran or electrolytes such as NaCl or CaCl2 (Hayat, 2000). To standardize our approach, we supplemented all fixatives, except the seawater, with 9% sucrose, according to the protocol from (Salvenmoser et al., (2010)).

Table 1 Detailed overview of the different fixation and washing buffer formulations and their osmolarity.

Buffer concentration	Buffer type	% Glutaraldehyde (vol/vol)	Addition	Mean osmolarity (mOsm)	s.d.	Comment	Abbreviation	
–	–	2.5%	–	287	±6.9			
–	Filtered seawater	–	–	1,100	±8.0			
–	Filtered seawater	2.5%	–	1,252	±15.2		FSW	
1X	PHEM buffer	–	–	219	±1.2			
1.5X	PHEM buffer	–	–	323	±1.4		–	
3X	PHEM buffer	2.5%	–	1,071	±6.9			
1.5X	PHEM buffer	2.5%	9% Sucrose	1,076	±1.6		marPHEM	
1.5X	PHEM buffer	–	9% Sucrose	714	±0.5	Washing solution		
0.1M	Phosphate buffer saline	–	–	300	n.d.			
0.1M	Phosphate buffer saline	2.5%	9% Sucrose	1,046	±2.9		marPBS	
0.1M	Phosphate buffer saline	–	9% Sucrose	645	±6.0	Washing solution		
0.1M	Sodium-cacodylate buffer	–	–	339	±4.5			
0.1M	Sodium-cacodylate buffer	2.5%	9% Sucrose	960	±2.9		marCaco	
0.1M	Sodium-cacodylate buffer	–	9% Sucrose	632	±5.8	Washing solution		

Osmolarity of seawater (salinity 35 PSU), buffer and fixative solutions was measured using either an Osmomat 030 (Gonotec, Berlin, Germany) or an Advanced Micro Osmometer Model 3MO Plus (Advanced Instruments, Norwood, MA, USA). All samples were tested in duplicate and measured independently three times. Mean values of sample readings were used for further calculations.

Sampling and specimen preparations

Mytilus edulis were obtained from a local fish market. They were transferred into an aquarium for 2 days to allow them to recover and to discard dead specimens. Three M. edulis were opened by cutting the adductor muscles and the gills were dissected. For each specimen, roughly equal-sized gill pieces were transferred into five different fixatives (Table 1). To avoid bias during the dissection, the tubes containing the fixatives were randomized before starting. Samples were fixed for 12 h at 4 °C and subsequently washed three times in their corresponding buffer solution (1.5X PHEM with 9% sucrose added, 0.1M cacodylate buffer with 9% sucrose added, 0.1M PBS with 9% sucrose added or filtered seawater) and post-fixed with 1% osmium tetroxide in ddH2O for 1 h. The samples were dehydrated in a graded ethanol series (30%, 50%, 70%, 100% twice), transferred into 100% dry acetone, and infiltrated using centrifugation (modified from McDonald, 2014) in 2 ml tubes sequentially with 25%, 50%, 75% and 2 × 100% Agar Low Viscosity resin (Agar Scientific, Stansted, Essex, United Kingdom). During this process, the samples were placed into the tube and centrifuged for 30 s with a bench top centrifuge (Heathrow Scientific, USA) at 2,000 g for each step. After the second pure resin step, they were transferred into fresh resin in embedding molds and polymerized at 60 °C in the oven for 24 h.

Bathymodiolus childressi were collected at 28°07′25.1″N 89°08′23.8″W at a depth of 1,071 m using the ROV Hercules in May 2015. Upon recovery, mussels were processed in chilled sea water. Specimen were fixed with PHEM buffered GA and embedded as described for Mytilus edulis.

Light and electron microscopy

Semi-thin (1 µm) and ultra-thin (70 nm) sections were cut with an Ultracut UC7 (Leica Microsystem, Wien, Austria). Semi-thin sections were transferred on a glass slide and dried on a heating plate at 60 °C. Sections were stained with 1% toluidine-blue solution (Sigma-Aldrich, St. Louis, MO, USA) for 20 s, rinsed three times with ddH2O then dried. A drop of LVR resin was placed on the slide, followed by a coverslip, and after polymerization, the sections were viewed using an Olympus BX 53 microscope (Olympus Corporation, Tokyo, Japan) and images were captured using a Canon EOS 700D camera (Canon Inc., Tokyo, Japan).

Ultra-thin sections were mounted on formvar coated slot grids (Agar Scientific, Stansted, Essex, United Kingdom) and contrasted with 0.5% aqueous uranyl acetate (Science Services, München, Germany) for 20 min and with 2% Reynold’s lead citrate for 6 min. Ultrathin sections were imaged at 20 kV with a Quanta FEG 250 scanning electron microscope (FEI Company, Hillsboro, OR, USA) equipped with a STEM detector using the xT microscope control software ver. 6.2.6.3123. Where needed, brightness and contrast of images was adjusted using Photoshop CS6 and figures were assembled using Adobe Illustrator CS6 (Adobe Systems, Inc., San Jose, CA, USA).

Results and Discussion

For the comparative part of the study, small pieces of the same gill were fixed in parallel with a set of fixatives to avoid sample bias. Samples from the same animal were always processed simultaneously (e.g., for embedding, polymerization, staining, etc.) to avoid handling bias. This procedure was repeated for three different animals to ensure reproducibility. To illustrate the differences between the individual fixative buffers, the result section shows representative images comparing the set of fixatives from the same animal.

Osmolarity measurements and buffer compositions

The osmolarity of 2.5% glutaraldehyde in ddH2O, sterile filtered seawater as well as of the working solutions of buffer and fixatives (Table 1) was measured. As the osmolarity of seawater was 1,100 mOsm, all fixative solutions were adjusted to be within a similar osmotic range with either sucrose or additional buffer concentrate. PBS and sodium cacodylate buffer were 300 and 339 mOsm respectively at a 0.1M concentration. PHEM buffer in its 1X concentration was only 219 mOsm, therefore, we increased the buffer concentration to 1.5X, resulting in an osmolarity of 323 mOsm, to be comparable with the other two buffers. An increase of the buffer to 3X concentration, to avoid the addition of sucrose in the fixative was measured, resulting in an osmolarity of 1,071 mOsm. In comparing the ultrastructure of 3X PHEM fixation and 1.5X PHEM + 9% sucrose, little difference was observed (Fig. S1); however, for the sake of clarity, we focused only on the sucrose adjusted fixatives.

The osmolarity of the fixative solutions ranged from 960 mOsm (Sodium cacodylate buffered GA), 1,046 (PBS buffered GA), 1,076 (PHEM buffered GA) to 1,252 (seawater—GA). For the sake of brevity, we use the following abbreviations throughout the rest of the text: Sodium cacodylate buffered GA (marCaco), PBS buffered GA (marPBS), PHEM buffered GA (marPHEM) and seawater buffered GA (FSW).

Comparing the effect of different buffers on the fixation of Mytilus edulis gill filaments

For the sake of clarity, images of the same regions of interest were taken from all samples prepared. To facilitate easy comparison, we focused on the typical organelles and structures expected in eukaryotic cells: mitochondria, nucleus, nuclear pores, Golgi apparatus, cilia, microvilli and rough ER. All images show transverse sections through gill filaments showing the ciliated frontal surface.

Light microscopy

At the light microscopic level, the variation in staining intensity infers a marked difference in tissue preservation between the different fixations (Fig. 1). The sections were stained with toluidine blue, a basic thiazine metachromatic dye which has a high affinity for acidic tissue components, including nucleoids, acidic mucus, RNA, etc. (Sridharan & Shankar, 2012). In the FSW fixed gill tissue (Fig. 1A) the outline of the nuclei, the cilia and the overall morphology are visible. By comparison, in the marCaco (Fig. 1B) fixed sample the nuclei are prominent, but the outline of the morphology is hard to detect. In the marPHEM (Fig. 1C) and marPBS (Fig. 1D) fixed samples, the cells are stained more strongly and homogenously with nuclei, cilia and a mucus cell easily discernible.

Figure 1 Light micrograph comparing four differently fixed tissue pieces of Mytilus edulis.

Overview of four differently fixed Mytilus edulisgills. All images show transverse sections through gill filaments focusing on the ciliated frontal surface. (A) shows a FSW fixed sample, (B) a marCaco fixed sample, (C) a marPHEM fixed sample and (D) a marPBS fixed sample. ci, cilia; mu, mucus granule; nu, nucleus.

Electron microscopy

Collectively, the direct comparison of the same gill tissue, fixed at the same time and processed identically, showed a pronounced disparity in terms of membrane contrast and retention of cytosol with different buffers, while larger ultrastructural organelles could be identified in all of them.

The FSW fixed samples showed a reasonable detail and preservation in the overview (Fig. 2A). The nuclear membranes were smooth but the chromatin was patchily distributed (Fig. 2B). The Golgi apparatus looked slightly collapsed and the cytosol was extracted. Due to the extracted cytosol, the rough ER stands out more prominently. Nuclear pores were visible (Fig. 2C). Both the mitochondria and the cilia were well preserved (Fig. 2D).

Figure 2 Ultrastructural details of FSW fixed gill cell of Mytilus edulis.

(A) shows an overview of the cells of the ciliated frontal surface, (B) is a higher magnification of the same, (C) shows the nucleus in higher magnification and (D) shows details of the cell surface. go, Golgi apparatus; nu, nucleus; ci, cilia; mu, mucus granule; mv, microvilli; np, nuclear pore; rer, rough ER; mi, mitochondria.

Figure 3 Ultrastructural details of marCaco fixed gill cell of Mytilus edulis.

(A) shows an overview of the cells of the ciliated frontal surface, (B) is a higher magnification of the same, (C) shows the nucleus in higher magnification and (D) shows details of the cell surface. go, Golgi apparatus, nu, nucleus; ci, cilia; mu, mucus granule; mv, microvilli; np, nuclear pore; cr, ciliary root; rer, rough ER; mi, mitochondria.

The marCaco fixed sample showed less overall contrast (Fig. 3A). Although nuclear membranes appeared parallel, with nuclear pores visible, the nuclei themselves look extracted and empty (Fig. 3B). The Golgi-Apparatus was well preserved and highly visible, however, the cytosol appeared extracted (Fig. 3C). The cilia were well preserved but the mitochondria appeared grainy and less electron-dense compared to the FSW fixed ones (Fig. 3D).

As already suggested by the light micrograph results, the marPHEM fixed sample showed noticeable improvements compared to the two previous samples (Fig. 4A). The nuclei were less extracted in contrast to the FSW and marCaco samples (Figs. 4B and 4C). The nuclear membranes were well defined and parallel with nuclear pores visible. The individual membranes of the stack of membranes of the Golgi-Apparatus could be easily discerned and the cytosol had overall a much more electron-dense appearance (Fig. 4C). Both microvilli and cilia were well defined and the mitochondria were more electron-dense than in samples fixed with the previous two fixative solutions (Fig. 4D).

Figure 4 Ultrastructural details of marPHEM fixed gill cell of Mytilus edulis.

(A) shows an overview of the cells of the ciliated frontal surface, (B) is a higher magnification of the same, (C) shows the nucleus in higher magnification and (D) shows details of the cell surface. go, Golgi apparatus; nu, nucleus; ci, cilia; mu, mucus granule; mv, microvilli; np, nuclear pore; cr, ciliary root; rer, rough ER; mi, mitochondria.

The marPBS fixed sample showed improved ultrastructural detail when compared to the FSW and marCaco fixed samples, similar to the marPHEM results (Fig. 5A). Membranes were well preserved and parallel, nuclear pores were visible and the cytosol had overall an electron dense appearance (Figs. 5B and 5C). However, the outline of many of the membrane stacks of the Golgi-apparatus (Fig. 5C) could not be so easily traced compared to the marPHEM sample. As in all other fixations, the cilia were well preserved and the mitochondria of the marPBS (Fig. 5D) appeared similarly well preserved as the marPHEM.

Figure 5 Ultrastructural details of marPBS fixed gill cell of Mytilus edulis.

(A) shows an overview of the cells of the ciliated frontal surface, (B) is a higher magnification of the same, (C) shows the nucleus in higher magnification and (D) shows details of the cell surface. go, Golgi apparatus; nu, nucleus; ci, cilia; mu, mucus granule; mv, microvilli; np, nuclear pore; cr, ciliary root; rer, rough ER; mi, mitochondria.

This experiment was designed to determine the influence the buffer has on the ultrastructural preservation. Some structures seemed unaffected by the change of buffer, for example cilia. As cilia structures are small and located on the surface of the tissue, they are also the first structures to come in contact with the fresh fixative and hence were well maintained with all methods. On the other hand, preservation of the nucleus and especially the cytosolic components varied strongly. Both marPHEM and marPBS showed generally better resolved ultrastructure than FSW and marCaco fixed tissue. The pronounced improvement in ultrastructural preservation observed in both marPHEM and marPBS suggest that they are a viable alternative for the fixation of marine invertebrates.

We were surprised by the comparatively poor performance of marCaco in these experiments, especially regarding the pronounced cytosol extraction we observed, considering it is one of the standard buffers for electron microscopy (Hayat, 2000; Kuo, 2014). Traditionally, the advantage of using cacodylate-based buffers over phosphate-based buffers was to avoid the formation of precipitates. However, no such precipitates were observed in this study. Looking at the literature, zwitterionic buffers like PIPES and HEPES are often remarked as “a class of buffers little used in electron microscopy” (Dykstra & Reuss, 2003) but at the same time it is mentioned that they might yield superior results, as they do not compromise elemental analysis and increase tissue retention (Baur & Stacey, 1977; Dykstra & Reuss, 2003; Griffith, 1993; Hayat, 2000; Kuo, 2014).

When working with buffer systems, one also has to consider the acid dissociation constant (pKa, here given at 20 °C). pKa indicates at which pH the buffer system is most effective to resist addition of either acid or base and has the highest buffering capacity. The optimal buffering region is usually considered to be around 1 pH unit on either side of the pKa. PBS (pKa = 7.21), HEPES (pKa = 7.55) and PIPES (pKa = 6.8) are much closer to the pH of the fixative (pH = 7.4), compared to sodium cacodylate (pKa = 6.27). This would imply that the buffering capacity of sodium cacodylate at pH 7.4 is reduced, compared to the other buffers, and might explain the difference in preservation.

Applying PHEM buffered fixation in the field

The only discernable difference between the marPHEM and marPBS was that the membrane contrast and definition was better in the marPHEM fixative. As our research on Bathymodiolus childressi requires excellent membrane definition, the marPHEM fixative was used for subsequent work. The gill filament (Fig. 6A) showed excellent preservation, both in the ciliated frontal part as well as in the region bearing the bacteriocytes. The close-up of the bacteriocyte (Fig. 6B) showed the distribution of the methanotrophic bacteria in the cell, with the typical enlarged lysosomes that have been reported in these cells, located basally. The high magnification of the symbiont (Fig. 6C) allowed us to clearly see the host membranes surrounding the symbiont, and the individual membranes of the intracytoplasmic membrane stacks, typical for type I methanotrophs, could clearly be discerned.

Figure 6 Ultrastructural detail of marPHEM fixed Bathymodiolus childressi gill filament.

(A) shows an overview of the ciliated frontal part of the gill filament, with the ciliated cells of the ciliated edge on the right hand side of the image and the symbiont containing bacteriocytes on the left hand side of the image. (B) shows a single bacteriocyte, containing the chemoautotrophic methane-oxidizer symbiont and the characteristic lysosomes. (C) shows a single symbiont surrounded by the hosts membrane, with the bacterial nucleoid, storage vacuoles and the typical methane-oxidizer membrane stacks clearly visible. ba, bacterium; bm, basal membrane; ci, cilium; hm, host membrane; icm, intracytoplasmatic membranes; ly, lysosomes; mu, mucus cell; mv, microvilli; nc, bacterial nucleoid; nu, nucleus; sv, storage vacuoles.

The current standard for ultrastructural fixation is high pressure freezing (HPF) followed by freeze substitution and resin embedding. However, for samples like the mussels of the genus Bathymodiolus, high-pressure freezing is not available. The mussels are retrieved from the seafloor from between 500 to 3,000 m water depth and cannot be cultivated in their natural state in the laboratory so far. Any HPF processing would need to happen at sea, on board marine research vessels. This is currently not possible, as HPF equipment is bulky, fragile and requires large volumes of liquid nitrogen. Therefore, optimizing the immersion fixation technique for excellent morphology was the focus of this study.

This experiment showed that we could replicate the excellent ultrastructural preservation results we obtained with marPHEM and Mytilus edulis, when applying it to the deep-sea mussel Bathymodiolus childressi.

Application of marPHEM fixation to a wider range of samples and additional information

Since the start of this study, marPHEM fixative has been used in our lab to investigate the ultrastructure of multiple marine invertebrates like Paracatenula galateia (Fig. S2), multiple nematodes of the sub-family Stilbonematinae (data not shown), the marine acoel Convolutriloba longifissura (Fig. S3), and the unicellular ciliate Kentrophorus sp. (Fig. S4). A more diluted formulation (2.5% GA in 0.06X PHEM) was used for investigating the terrestrial soil archaea Nitrososphaera viennensis (Stieglmeier et al., 2014) as well as E. coli (Montanaro et al., 2015) and a 1X PHEM formulation without sucrose was used for the fixation of the limnic flatworm Stenostomum cf. leucops. (Fig. S5).

Experience showed that the 10X PHEM stock solution can be stored frozen at −20 °C for at least a year without any obvious detrimental effect. Depending on the concentration used, the liter pricing of PHEM buffer is in the same range as cacodylate buffer, but usage of PHEM buffer results in less hazardous waste being produced.

Conclusions

This study compares the influence of different buffers on the resulting ultrastructural morphology preservation and demonstrates the effectiveness of the isosmotic, non-toxic PHEM buffer in combination with aldehydes when applied as an immersion fixative. We have adapted this buffer-fixative combination for ultrastructural fixation of marine invertebrates. The individual components of PHEM buffer seem to enhance ultrastructural detail, reduce extraction and preserve membrane integrity. Our samples showed no evidence of shrinkage, excellent structural preservation and, due to their contrast, easily discernable membranes. As Hayat (2000) already stated “no single buffer can claim universal superiority over the others.” We therefore acknowledge that our findings might not be transferrable to every sample, but we would like to encourage other researchers to use marPHEM for two reasons: (a) The above results showcase how changing the buffer in an EM fixative can result in substantial improvements in ultrastructure. Therefore, obtaining satisfying ultrastructural preservation might be as simple as switching to a non-toxic buffer like marPHEM or marPBS. (b) Replacement of toxic solutions with non-toxic alternatives in electron microscopy samples preparation protects both the researchers’ health and reduces toxic environmental waste. Even if the results, when switching from sodium cacodylate to e.g., marPHEM, are only on par, the simple fact that the handling and disposal of one more toxic chemical can be eliminated, should provide sufficient motivation.

Taken together, our comparative studies showed that isotonic PHEM buffered fixation (marPHEM) gave equal or better fixation and subsequent ultrastructural preservation when directly compared to conventional fixatives. We highly recommend PHEM buffer with glutaraldehyde as an electron microscopy fixative solution for routine use.

Supplemental Information

Figure S1 Comparison of two different fixatives, using Mytilus edulis gill tissue

Fixative containing 3X PHEM and 2.5% GA (A, C, E) and fixative containing 1.5X PHEM, 2.5% GA and 9% sucrose (B, D, E). (A) and (B) show a light microscopic overview of the ciliated frontal surface. (C) and (D) are a close-up of the nucleus and Golgi apparatus and (E) and (F) show details of the cell surface. nu, nucleus; ci, cilia; go, Golgi apparatus; mi, mitochondria.

Click here for additional data file.

Figure S2 Ultrastructural detail of Paracatenula galateia

Transmission electron micrographs of Paracatenula galateia (A) is part of a cross section, showing the location of the bacteria in the bacteriocytes and (B) is a high magnification of the epidermal cell layer and the underlying bacteria. The epidermis is overlaid by a thick layer of extracellular mucus. ba, bacteria; ep, epidermis; em, extracellular mucus; nu, nucleus; ci, cilia; mu, mucus granule.

Click here for additional data file.

Figure S3 Ultrastructural detail of Convolutriloba longifissura

Transmission electron micrographs of Convolutriloba longifissura (A) shows part of a nucleus and multiple golgi, (B) shows the symbiotic algae, (C) shows the sagittocyst and (D) is a higher magnification of the symbiotic algae. Arrowheads indicate Golgi complexes in (A), chloroplast membrane stacks in (B), desmosomes in (C) and flagellal basal bodies in (D).

Click here for additional data file.

Figure S4 Ultrastructural detail of Kentrophoros sp

Transmission electron micrograph of Kentrophoros sp. Fig. S4 shows a cross section of the unicellular ciliate Kentrophoros, which is associated with symbiotic bacteria. ci, cilia; cr, ciliary roots; ba, bacteria; fv, food vacuole; mi, mitochondria.

Click here for additional data file.

Figure S5 Ultrastructural detail of Stenostomum cf. leucops

Transmission electron micrographs of Stenostomum cf. leucops (A) shows a cross section of the catenulid flatworm, with the gut lumen, and gut epithelium clearly visible. (B) is a close-up of the protonephridial duct, with multiple cilia being visible. gl, gut lumen; ge, gut epithelium; ci, cilia; nu, nucleus; mi, mitochondria; ms, muscles.

Click here for additional data file.

Supplemental Information 1 Supplementary methods

Click here for additional data file.

We are very grateful to the Core Facility Cell Imaging and Ultrastructure Research of the University of Vienna for technical support, to D Abed-Navandi from the public aquarium Haus des Meeres aquarium and B Egger for providing samples. The authors wish to thank HR Gruber-Vodicka for editorial advice. The authors would like to thank the Ocean Exploration Trust and the captain and crew of the E/V Nautilus for their support. This work is contribution 987 from the Carrie Bow Cay Laboratory, Caribbean Coral Reef Ecosystem Program, National Museum of Natural History, Washington, D.C.

Additional Information and Declarations

Competing Interests

Author Contributions

Field Study Permissions

Data Availability

The authors declare there are no competing interests.

Jacqueline Montanaro conceived and designed the experiments, performed the experiments, analyzed the data, contributed reagents/materials/analysis tools, wrote the paper, reviewed drafts of the paper.

Daniela Gruber analyzed the data, contributed reagents/materials/analysis tools, reviewed drafts of the paper.

Nikolaus Leisch conceived and designed the experiments, performed the experiments, analyzed the data, contributed reagents/materials/analysis tools, wrote the paper, prepared figures and/or tables, reviewed drafts of the paper.

The following information was supplied relating to field study approvals (i.e., approving body and any reference numbers):

Permission for the collection and export of invertebrate animals (only applicable to Fig. S2) was issued by the Ministry of Forestry, Fisheries and Sustainable Development of Belize.

The following information was supplied regarding data availability:

DOI 10.6084/m9.figshare.2060718.

Figshare: https://figshare.com/s/4816561571c6366077dd.

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
