# Peer review of "Improved ultrastructure of marine invertebrates using non-toxic buffers"

_PeerJ, doi:10.7717/peerj.1860_

## Round 0.1 · original submission · Minor Revisions

· Academic Editor

Minor Revisions

Both reviewers were very positive in their reviews of your manuscript, only suggesting minor changes. I have added a few additional editorial comments, and thus there are two attachments and a few comments for you to respond to. As well, I have noticed here and there the English needs a bit of work (see my attachment), please ensure that this is gone over very thoroughly before any resubmission. Based on the positive reviews and generally small corrections, my decision is "minor revision".

·

Basic reporting

Authors report on the evaluation of preparatory methods used in marine organisms microscopy, speciffically on different buffers implementation in fixation process. The aim is to optimize ultrastructure preservation using methods avilable in field work on ships (sampling in remote locations) and by procedures that omit toxic chemicals as much as possible. They provide clear and sufficient introduction and refer to relevant literature.
Figures are of high quality and demostrate the statements in the text. The efforts to optimize samples preparation in microscopy with respect to minimize the toxic impacts of chemicals necessary are valuable for the community and carefull evaluations like this study are a welcome contribution to a broader field of microscopy.

Experimental design

The motivation, background and general framework of the study are presented in a clear and complete manner. The aim of the study is clearly defined and the results reported in the manuscript are answering the presented questions and are discussed thoroughly. Methodologic procedures are described very precisely.

Validity of the findings

The manuscript includes a sufficient number of micrographs that are of high quality and support the conclusions in the text. Conclusions are clear and the results are extensively discussed, including relevant relation to literature data.

Additional comments

The study is conducted carefully and presented in a concise and clear way.
Minor comments/suggestions are included in the attached manuscript file.
Concerning figure captions:
Figs. 2,3,4,5 : maybe better use 'cells' in the title, instead of 'cell';
2B, 3B, 4B, 5B: ..is a higher magnification of the apical...
in general - using arrows at labels (where appropriate) would make labeling more clear for a reader not so familiar with the content;
Fig. 5: 'cr' is in the caption, but not used as a label on the figure.

·

Basic reporting

This article is very clearly written. The introduction presents the motivation for the study, and emphasizes the importance of proper fixation of invertebrate tissues, along with the challenges of working in the field or at sea. The authors adequately compare the most commonly used fixatives and buffers, and justify their reasoning for pursuing a non-toxic buffer for use in conjunction with glutaraldehyde fixation.

Figures are entirely relevant and of very high quality. Images clearly show the ultrastructural differences that resulted from different fixation protocols. Supplementary information presents micrographs of other invertebrates prepared using the new, PHEM-buffered fixation protocol, helping to demonstate that this technique can be used with excellent results for studying different taxa (including non-marine species).

Minor comments:
line 99: "Mytilus" should be written in its entirety.
line 110-111: add spaces between numbers and units (mM)
line 127: "dead specimens"
line 141: add space: "1071 m"
line 147: add space: "1 um"
line 199: "... morphology are visible."
line 200: "... in the marCaco..."
line 245: "...buffers for electron..."
line 268: "...the typical enlarged lysosomes..."
line 275: "3000 m"

Experimental design

The authors clearly defined the research question and justify how the study fills a knowledge gap. Authors provided detailed information on their methodology that ensure reproducibility by other investigators.

line 158-159: what exactly is the image processing to which you refer?

Validity of the findings

The findings are robust: the images presented (replicate tissue samples processed from the same individuals, but using different fixation treatments) clearly show the advantages of using the PHEM or PBS buffered fixatives over a more toxic alternative that is more commonly used (sodium cacodylate buffered fixative), and over fixation using filtered seawater as a buffer. PHEM-buffered fixation produces similar results to PBS-buffered fixation, but membrane preservation is slightly improved using the former technique.
This study presents researchers with attractive, non-toxic alternatives to traditionally used electron microscopy buffers.

Additional comments

This article was a real pleasure to read. I found it clear, informative, and well referenced. I appreciated the table presenting the osmolarity of different solutions, and how the figures allowed the reader to directly compare the results of different fixation procedures. Like the authors, I was surprised by the outcome, and am now convinced that non-toxic buffers are the way to go with marine invertebrate sample preservation for ultrastructural work.

---

## Round 0.2 · accepted · Accept

· Academic Editor

Accept

The authors have revised their submission well, answering all comments and questions as needed. There are no outstanding issues with regard to this manuscript, and therefore I have reached the decision of 'accept'.